# Reverse Phase Protein Array Profiling Identifies Recurrent Protein Expression Patterns of DNA Damage-Related Proteins across Acute and Chronic Leukemia: Samples from Adults and the Children’s Oncology Group

**DOI:** 10.3390/ijms24065460

**Published:** 2023-03-13

**Authors:** Fieke W. Hoff, Ti’ara L. Griffen, Brandon D. Brown, Terzah M. Horton, Jan Burger, William Wierda, Stefan E. Hubner, Yihua Qiu, Steven M. Kornblau

**Affiliations:** 1Department of Internal Medicine, UT Southwestern Medical Center, Dallas, TX 75390-9030, USA; 2Department of Microbiology, Biochemistry and Immunology, Morehouse School of Medicine, Atlanta, GA 30310-1458, USA; 3Division of Pediatrics, The University of Texas M.D. Anderson Cancer Center, Houston, TX 77030-4009, USA; 4Department of Pediatrics, Texas Children’s Cancer Center, Baylor College of Medicine, Houston, TX 77030-3498, USA; 5Department of Leukemia, The University of Texas M.D. Anderson Cancer Center, Houston, TX 77030-4009, USA

**Keywords:** leukemia, RPPA, DNA damage, proteomics

## Abstract

DNA damage response (DNADR) recognition and repair (DDR) pathways affect carcinogenesis and therapy responsiveness in cancers, including leukemia. We measured protein expression levels of 16 DNADR and DDR proteins using the Reverse Phase Protein Array methodology in acute myeloid (AML) (*n* = 1310), T-cell acute lymphoblastic leukemia (T-ALL) (*n* = 361) and chronic lymphocytic leukemia (CLL) (*n* = 795) cases. Clustering analysis identified five protein expression clusters; three were unique compared to normal CD34+ cells. Individual protein expression differed by disease for 14/16 proteins, with five highest in CLL and nine in T-ALL, and by age in T-ALL and AML (six and eleven proteins, respectively), but not CLL (*n* = 0). Most (96%) of the CLL cases clustered in one cluster; the other 4% were characterized by higher frequencies of deletion 13q and 17p, and fared poorly (*p* < 0.001). T-ALL predominated in C1 and AML in C5, but both occurred in all four acute-dominated clusters. Protein clusters showed similar implications for survival and remission duration in pediatric and adult T-ALL and AML populations, with C5 doing best in all. In summary, DNADR and DDR protein expression was abnormal in leukemia and formed recurrent clusters that were shared across the leukemias with shared prognostic implications across diseases, and individual proteins showed age- and disease-related differences.

## 1. Introduction

The leukemias are malignancies arising from the white blood cell stem cell population that, based on the lineage of the cell of origin (myeloid vs. lymphoid) and the speed of presentation (acute vs. chronic), are broadly categorized into four groups. The annual incidence of all forms of leukemia is approximately 60,000 newly diagnosed cases across the United States. With the introduction of tyrosine kinase inhibitors that target the BCR-ABL fusion gene in chronic myeloid leukemia (CML), and with the introduction of agents targeting Bruton tyrosine kinase and their associated pathways in chronic lymphocytic leukemia (CLL), significant improvements have been made in survival of chronic leukemia. However, despite the recognition of the genetic, epigenetic and mutational landscape in acute leukemia, most of the involved genes are not yet specifically targetable and most acute leukemia patients die of their disease, accounting for the majority of the 24,000 annual deaths from leukemia [1,2,3,4]. Thus, better understanding of leukemogenesis and recognition of new targetable proteins are required to improve outcome.

As presented elsewhere in this Special Issue, DNA damage recognition (DNADR) response and repair (DDR) pathways are involved in the pathogenesis of many types of cancer, including leukemia. DDR can modulate the sensitivity to initial treatment and can later become a mechanism of resistance. Numerous pathways with multiple protein components affect the functionality of the various pathways controlling DDR. Some of these genes are mutated in some forms of cancer, but within leukemia, none of these genes are mutated in more than 1% of the cases [5,6,7,8,9]. Thus, within leukemia, it may be the relative expression levels and activity of DNADR and DDR pathway proteins that modulate the functional level of DDR repair. However, the expression level and activity of these proteins has rarely been studied in leukemia, and a comprehensive simultaneous analysis of multiple DNADR and DDR proteins has not been reported. We hypothesized that there would be recurrent patterns of DDR protein expression found in different leukemias and that these would be prognostic of outcome, as well as identifying possible therapeutic targets for intervention on a personalized basis.

We have used Reverse Phase Protein Array (RPPA) technology to study protein expression in a large series of acute myeloid leukemia (AML) and T-cell acute lymphoblastic leukemia (T-ALL) in both adult and pediatric populations, as well as CLL in an adult population [10,11]. In RPPA whole cell protein lysates, prepared fresh from leukemia enriched bone marrow or peripheral blood samples, are printed in serial dilutions onto slides (up to 1152 samples per slide) and each slide is then probed with a highly validated antibody against total proteins or post-translationally modified ((PTM) protein sites, including phosphorylation sites, histone methylation and acetylation sites and cleavage forms)) protein [12,13]. An overview of the number of patient samples and number of antibodies utilized on the different arrays is presented in Table 1. The accompanying Figure 1 shows which DDR protein-detecting antibodies were used in each of these arrays, as all were not used in every array. In this manner, a dataset of multiple proteins that form a large cohort of leukemia patients can be assembled. Our group has also developed computational methodology to recognize recurrent patterns of expression of all studied proteins collectively, as well as within more limited sets of functionally related proteins and individual proteins.

Many antibodies for detecting DNADR and DDR proteins were included in these datasets. This provides a heretofore unseen opportunity to look at the collective expression of DDR proteins in AML, CLL and T-ALL, as presented in this and the accompanying manuscripts, and to compare the expression patterns across the different leukemias, which is the focus of this manuscript. Specifically, we asked if DDR proteins had disease-specific protein expression patterns or if these were shared across diseases. We also analyzed AML and T-ALL to determine if there were age-specific differences and similarities between pediatric and adult populations. We also wondered if the same pattern would have the same prognostic implications across diseases.

## 2. Results

### 2.1. DDR Protein Expression Patterns Are Differentially Expressed between Acute and Chronic Leukemia Subtypes

We performed clustering analysis across the 2466 samples for 16 proteins and protein modifications (Figure 2), with protein expression normalized against expression of the same protein in normal bone marrow-derived CD34+ cells. Using the progeny clustering algorithm [14], we found an optimal number of five protein clusters that shared similar, correlated protein expression patterns. Acute leukemia clearly separated from CLL (Figure 2, cluster 2 (pink)). A strong majority of CLL cases (96%) clustered in C2 with rare cases found in clusters 3 (yellow), 4 (light green) and 5 (dark green). AML and T-ALL were both found in clusters 1, 3, 4 and 5, but cluster 1 was disproportionately high in T-ALL cases, while cluster 5 had a low percentage of T-ALL cases and was almost exclusively formed by AML. Of note, within two of the four clusters (Cluster 3 and 4), the adult AML patients (defined as age ≥ 18) clustered apart from the pediatric AML cases, with some patients intermixed (Appendix A, top annotation). Note the separation of most adult (blue) cases from pediatric (pink) cases, but the mixture of some “pediatric-like” adult cases among the pediatric cases. Within these four “acute-dominated” clusters, the T-ALL and AML cases did not separate from each other. Thus, AML and T-ALL share commonalities of DDR expression based on the main drivers of the cluster formation, but AML has a subset of patients with a DDR protein expression profile unique to AML.

We also assessed for global differences in expression between the three diseases as shown in Figure 3. We identified three distinct patterns. Level of significance was set at *p* < 0.05/16 = 0.003 after correction for multiple comparisons. The first pattern included proteins with expression that was highest in CLL and lower in T-ALL, with AML either lower or equal to the T-ALL cases. In this category were CHEK2, RAD50, RPA32, WEE1 and XPA. The second pattern has proteins with the highest expression in T-ALL, with AML mostly somewhat lower, and CLL either lower than or equal to the AML cases. In this category were CHEK1, CHEK1-pSer435, CHEK2-pThr68, MSH2, MSH6, RAD51, RPA32-pSer4_8, SSBP2 and XRCC1. Another two proteins were relatively equal across the three diseases, including PCNA and TP53. In combination, this is suggestive of wide differences between the three diseases, but indicates that they share common activation patterns.

### 2.2. DDR Protein Expression Levels Are Different across Age Groups in Acute Leukemia

Similarly, we looked for differences between different age groups within each of the three diseases (Appendix A). Within AML, nine proteins had a pattern in which levels were higher in the pediatric subgroups (ages <2, 2–10, 10–18 years), with two proteins (RAD51, WEE1) dropping in a linear fashion across all age groups and the other six progressively declining in the 18–29, 30–59 and 60+ subgroups starting after age 18 (CHEK2, CHEK2-pThr68, MSH2, RPA32, RPA32-pSer4_8, and XRCC1). Two other proteins (RAD50 and SSBP2) were very low in infant leukemia (<2 years), but then relatively similar across other age subsets with a tendency to drop again after the age of 60 years. Another five proteins showed no difference in expression across age groups (CHEK1, MSH6, PCNA, TP53 and XPA. In T-ALL, three proteins were higher in the younger age groups and declined with age (PCNA, RAD51, RPA32-pSer4_8) with RPA32-pSer4_8 and PCNA having a similar pattern in AML. Different from AML, three proteins increased with age (CHEK2-pThr68, SSBP2, XPA) and ten showed no association with age (CHEK1, CHEK1-pSer345, CHEK2, MSH2, MSH6, RAD50, RPA32, TP53, WEE1 and XRCC1). In CLL, predominantly a disease of older adults, none of the proteins were significantly different between patients < 60 years of age compared to those older than 60 at time of diagnosis. The pediatric population (35% of all AML cases) comprised only 11% and 16% of C2 and C5, respectively. With half of the DDR proteins showing age-related variation in AML and T-ALL, we conclude that DDR activity is different in pediatric and adult acute leukemia.

### 2.3. Disease Specific Characteristics Significantly Associated with DDR Protein Expression Patterns

We next examined whether certain clinical features and genetic events were associated with specific DDR protein signatures in each disease (Table 2, Table 3 and Table 4). Neither disease showed an imbalance in gender or race and ethnicity. While age was not imbalanced in T-ALL or CLL, age was strongly associated with cluster membership in AML (*p* < 0.001). In AML, infants (age < 2 years) were overrepresented in C1 (10% vs. 5% overall), and the 10–18-year-old patients were more frequently present in C1 (25%) and C3 (25%) but absent from C2 (vs. 18% overall). In contrast, the older 60+ cohort dominated C2 (61%) and C5 (59%) compared to 41% in the total cohort (*p* < 0.001).

Among the disease-specific characteristics, many imbalances were observed. In AML, a WBC over 100,000 was associated with C1 (*p* < 0.001), CNS leukemia was more common in the pediatric-dominated C1 and C3 (*p* < 0.001) and favorable risk group with C3 (*p* < 0.001). Similarly, many mutations were unevenly distributed with C3 having high proportions of NPM1, CEBPA and FLT3-ITD mutations (*p* < 0.001, *p* < 0.001, *p* = 0.001, respectively), as were cytogenetic events (*p* < 0.001). In T-ALL, all of the patients in the CLL-progeny clustering C2 (*n* = 2) were intermediate risk, not Hispanic or Latino, aged 2–10 years, had leukocytosis, and did not have CNS involvement. Other biases were a high proportion of Hispanic patients in C5 (43% vs. 22% overall, *p* = 0.05), and a low percentage of hyperleukocytosis cases in C4 (28%, *p* = 0.003)). As there were only 29 of 795 CLL patients who did not group into C2, these were grouped together and compared to C2-CLL. This group (black, “not CLL” in Table 4) was notable for a lower percentage of cases with a deletion of chromosome 13q, but a high percentage of cases with deletion 17p (TP53) (24% vs. 8%, *p* = 0.006) and chromosome 9 (10% vs. 2%, *p* = 0.003).

### 2.4. Expression Patterns Associated with Survival Outcome across Diseases

We further queried whether cluster membership as shown in Figure 2 was prognostic within each disease for outcome measures, and whether cluster membership had a similar implication across the different diseases, as shown in Figure 4. Within AML, cluster membership was not prognostic for overall survival (OS) in both pediatric (treated with cytarabine, daunorubicin, etoposide- [ADE] based protocols) and adult AML patients (treated under a variety of protocols (Appendix A)), but there was significant splay in the complete remission (CR) duration curves. For CR duration, prognosis was best for C5, followed in order by C3, C4 and C1, with a similar non-significant trend seen in pediatric AML cases. The CLL-dominated C2 had discordant prognostic impact, but the very small number of cases (pediatric, *n* = 3, adult *n* = 27) makes consideration of this cluster tenuous. The implications of DDR protein expression in CLL are analyzed in more detail in the accompanying manuscript.

In the T-ALL cluster, membership was prognostic in adults for both OS and CR duration, with C5 doing best followed by C1, C4 and C3. In pediatric T-ALL, C5 cases, albeit few in number, had 100% OS and EFS (*n* = 4), and while the other curves were tightly clustered, there was a similar hierarchy as seen in adults of C1 > C5 > C3. In CLL, membership in anything other than C2 conferred a dismal OS (*p* < 0.001). When comparing the hierarchy or response within acute leukemia cluster, membership had a similar prognosis across the two diseases, with best prognosis associated with C5, and C3 and C4 having a similarly poor outcome in all analyses. However, the pediatric-dominated C1 had a similar prognosis to C3 and C4 in pediatric leukemia, but a worse prognosis in adult AML.

## 3. Discussion

The existence of four separate proteomic datasets from pediatric AML, adult AML, pediatric and adult T-ALL and CLL, all analyzed using the same RPPA methodology in the same laboratory, provided us with a unique opportunity to compare DDR protein expression across different types of leukemia with the goal of identifying commonalities and disparities. Although RPPAs that were generated more recently included more DDR-related proteins than those developed earlier, 16 proteins or protein modifications were shared across all four arrays.

Comparing expression patterns, the first observation that was made is that there are five dominant, recurrent DDR protein expression patterns found across these three types of leukemia. Next, we found that CLL has a very distinct and unique signature (C2) that included 96% of the CLL patients, and that adult and pediatric AML and T-ALL shared four different expression signatures. This suggests that there are different mechanisms of DDR activity between chronic and acute leukemias. However, within the two acute leukemias, DDR patterns are shared across diseases. This implies that therapeutic interventions directed at particular expression levels may be applicable across leukemia types. It also raises the intriguing question of whether a similar set of restricted DDR protein expression patterns would be seen in other malignancies as well, including lymphomas and solid tumors, and whether these patterns will be shared across diseases. If so, then successful use of a DDR-targeting agent in one disease may also work in other diseases with a similar protein expression pattern.

We also observed that while the four acute leukemia-dominated clusters all had both adult and pediatric cases, protein cluster 1 was pediatric-dominant while C5 was predominantly formed by adult leukemia, and that within all four of these clusters there were adult-only regions and pediatric-dominated regions with some admixed adult cases. As the individual box plots show (Appendix A), within AML and T-ALL, about half of the proteins had age-dependent expression proteins. Thus, while adult and pediatric acute leukemia cases shared the dominant drivers of cluster recognition, there were age-dependent differences.

Protein cluster 1 was particularly interesting, as it had more proteins than the other clusters with protein expression higher than expressed in the healthy, non-leukemic CD34+ samples, most notably the bottom six proteins (lower dendrogram branch in Figure 2A), including CHEK1, both MSH2 and MSH6, RAD51, SSBP2 and WEE1, as well as being the only cluster with relatively higher levels of all three PTM-protein states (= activated DDR) forms of CHEK1, CHEK2 and RPA32. In both adult and pediatric AML, cluster 1 had poorer OS and the shortest CR durations. We hypothesize that this pattern may correspond to the highest DDR activity and consequently to greater repair capacity by these leukemic cells, subsequently resulting in more resistance to therapy. While this requires laboratory validation, it could suggest that this group is more likely to respond to broad targeting of DDR proteins. By comparison, C5, which was characterized by the highest expression of XPA (involved with NER) and TP53; relatively lower expression of the PTM forms; and very low expression of all of the proteins that are high in C1, was universally the best prognostically, in both pediatric and adult AML and T-ALL. This suggests that this group should not be treated with agents targeting the DDR components as they already do very well with conventional chemotherapy. Interestingly, in C3 and C4, with proportional mixes of adult and pediatric cases, AML and T-ALL have similar outcomes in both diseases and age groups. They are also characterized by higher expression of the three PTM proteins, suggesting DDR activation, and spotlighting these patients for agents targeting DDR activation through the CHEK and RPA proteins.

Within CLL, there was an interesting finding that 4% of cases had an atypical pattern that performed very poorly compared to other CLL cases, with a shorter time to first and second treatment [11] and markedly inferior survival. These patients were characterized by a higher percentage of adverse cytogenetic changes 13q and 17p (TP53). Further study of this small subset may elucidate pathophysiological differences underlying the different DDR reliance and responsiveness. There is a small subset of CLL cases that do not respond as well to modern BTK and PI3K inhibitors and perhaps this might identify the cause of their poorer response.

In summary, DDR protein expression shows heterogeneity in expression between chronic and acute leukemia, but commonalities between T-ALL and AML. Most pediatric and some adult patients share expression profiles, but there are some adult predominant patterns, suggesting that there are age differences. The combined DDR protein signatures have prognostic implications, but these are much better defined in the accompanying manuscripts in this issue which individually consider CLL and AML. These findings have implications for selection of therapy directed at DDR targets.

## 4. Materials and Methods

Full descriptions of the patients included in these datasets, RPPA methodology and computational methodology can be found in the accompanying manuscripts on DDR expression in AML and DDR expression in CLL.

Briefly, RPPA was performed on 2466 patients with AML (*n* = 1310), CLL (*n* = 795), T-ALL (*n* = 361) and 51 normal CD34+ samples from healthy subjects. Informed consent was obtained and the study was conducted in accordance with the Declaration of Helsinki. Collection and analysis of samples were in accordance with protocols approved by the MD Anderson Cancer Center investigational review board (Lab 01-473, Lab 03-0893, Lab 04-0678, Lab 08-0431, Lab 05-0654, Lab 07-0719), by local investigational review boards as per institutional requirements or as part of the Children’s Oncology Group (COG) trials (AAML1031 #NCT10371981, AALL1231 #NCT02112916) [15,16]. COG specimens were collected to address a protocol-specified aim to examine for changes in protein expression patterns using reverse phase protein lysate array. An overview with the most important patient characteristics is shown in Table 2, Table 3 and Table 4. Patients were treated under a variety of treatment protocols (Appendix A).

Within each of the separate arrays, different numbers of DDR proteins were examined, but 16 antibodies, against 13 individual DDR proteins (CHEK1, CHEK2, MSH2, MSH6, PCNA, RAD50, RAD51 RPA32, SSBP2, TP53, WEE1, XPA and XRCC1, as well as PTM forms of CHEK1, phosphorylated on serine 345 (CHEK1-pSer345), CHEK2 phosphorylated on threonine 68 (CHEK2-pThr68) and RPA32 phosphorylated on serine 4 and 8 (RPA32-pSer4_8), all of which induce DDR repair activity, were included on all the arrays. These 16 DDR-related proteins are among 540 antibodies that have been validated for use in RPPA, and were selected to provide a broad coverage of different pathways and cellular processes with an emphasis on genes shown to be involved in cancer pathogenesis, therapy resistance or prognosis. Not all potential targets have validatable antibodies. Thus, this list is directed, rather than all-inclusive. There are many different naming conventions for proteins in use, creating much confusion. To clarify which DDR proteins were studied in these three companion manuscripts, a “Rosetta stone” table listing the manufacturer, RPPA antibody name, HUGO protein abbreviation and the full name (from GeneCards) is presented along with RPPA staining details. In addition, Pearson correlation coefficients for validation, and primary and secondary antibody dilution can be found in Appendix A.

### Statistical Analysis

Replicates-based normalization [17], which requires at least 30 samples that are printed on both slides, was used to align samples from the four different slides. To set the median of the normal CD34+ samples at zero, expression levels for all samples were subtracted by the median of the 51 normal CD34+ sample. K-means [18] coupled with the progeny clustering algorithm [14] was applied to the protein expression data to identify an optimal number of protein clusters (i.e., patient subgroups with a similar correlated protein expression profile within each PFG). Principal component analysis was applied to graphically compare patients’ protein cluster expression patterns to those of non-malignant CD34+ cells. Survival curves for the five protein clusters were generated using the Kaplan–Meier estimator. Associations between protein clusters and clinical variables were assessed using the Fisher exact test for categorical variables and the Kruskal–Wallis test by ranks for continuous variables. All statistical analyses were performed using R Version 4.2.2 (2022-10-31. RStudio, Inc., Boston, MA, USA). Statistical analysis was not performed by the COG. 

## Figures and Tables

**Figure 1 ijms-24-05460-f001:**
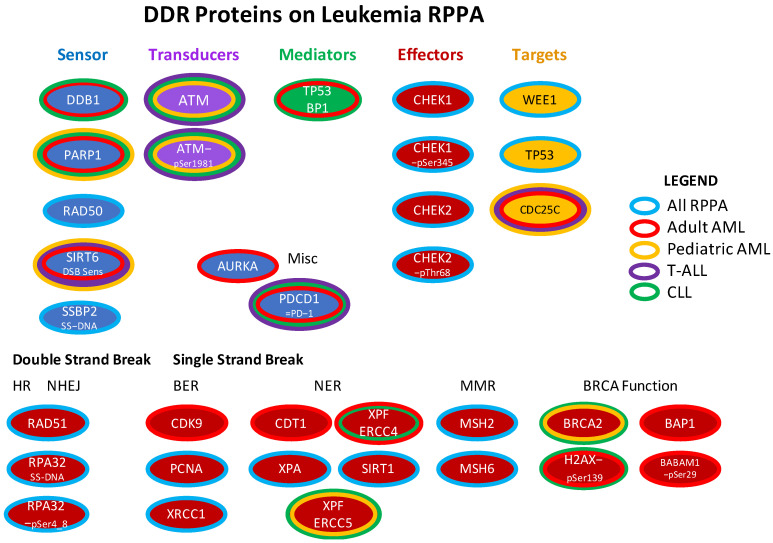
DDR proteins included in the four leukemia RPPA arrays. The DNADR and DDR-related proteins that were assessed in the four RPPAs (adult AML, pediatric AML, T-ALL, CLL) are shown arranged as per the organization in Figure 1 of Esposito and So [5], or classified by whether they are effectors of double or single strand break repair. Center color reflects the function as specified below the title. The surrounding band(s) show which array(s) they were printed on: blue [on all four arrays], red [adult AML], yellow [pediatric AML], green [CLL]. None of the antibodies were unique to the T-ALL RPPA. Repair mechanism abbreviations: HR = homologous recombination. NHEJ = non-homologous end joining. BER = breakpoint excision repair, NER = nucleotide excision repair. MMR = mismatch repair. Figure adapted from Esposito (2014).

**Figure 2 ijms-24-05460-f002:**
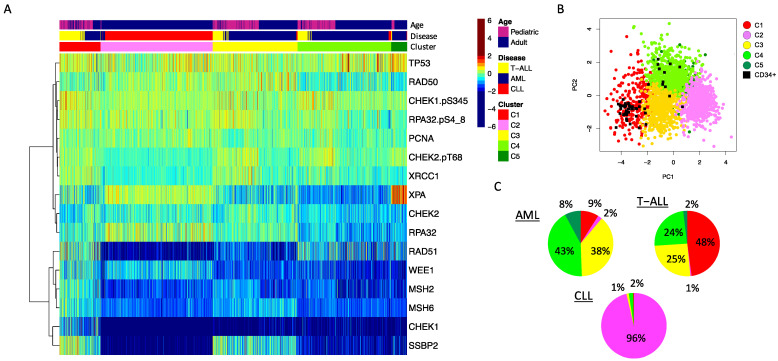
DDR protein expression in leukemia. (**A**) The heatmap shows protein expression scaled relative to normal bone marrow-derived CD34+ cells (scaled from high = burgundy, normal = green, low = blue) in 2466 leukemia patient samples. Annotations show: top row: age (pink = pediatric (age < 18 years), blue = adult (age ≥ 18 years)), middle row: disease: blue = AML, red = CLL, yellow = T-ALL. Unbiased hierarchical clustering followed by progeny clustering defined five different clusters, shown in the third row of the annotations. (**B**) The principal component graph shows that the normal CD34+ samples, shown as a black square, overlap with protein cluster 1 (red) and protein cluster 4 (green). (**C**) Pie charts showing the distribution of cluster membership for the AML, T-ALL and CLL patient samples. Colors align with the colors used in the annotation bar of the heatmap shown in A.

**Figure 3 ijms-24-05460-f003:**
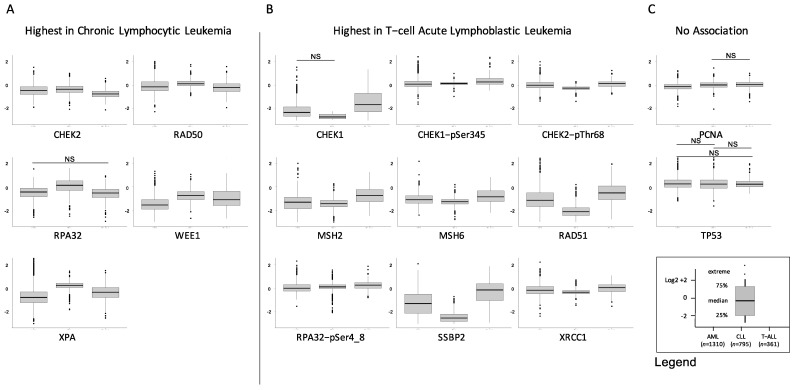
Differential expression of DDR proteins by disease. Individual box plots comparing the expression of AML (left, *n* = 1310), CLL (center, *n* = 795) and T-ALL (right, *n* = 361) are shown for the 16 DDR proteins. Most proteins fell into one of three patterns: (**A**) Highest expression in CLL, with expression in T-ALL either above or equal to that of AML (*n* = 5 proteins), (**B**) Highest expression in T-ALL, followed by lower expression in AML, and then by expression in CLL that was either lower or equal (*n* = 9 proteins), (**C**) No association with disease (*n* = 2), NS = not significant.

**Figure 4 ijms-24-05460-f004:**
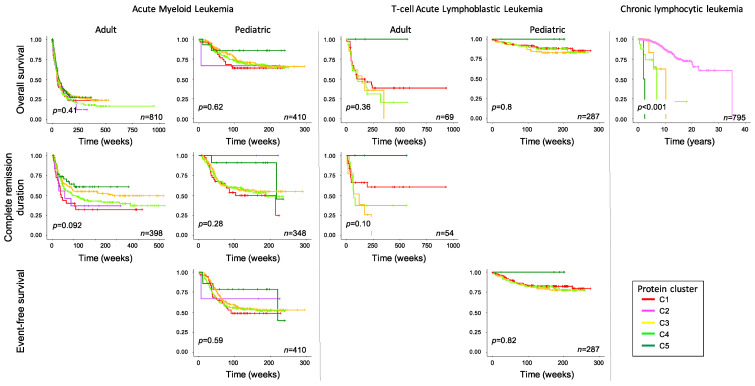
Association of DDR cluster membership with survival outcome. Overall survival (top row), complete remission duration (middle row) and event-free survival (bottom row) are shown for acute myeloid leukemia (left), T-cell acute lymphoblastic leukemia (middle), both stratified into adult and pediatric age groups, and chronic lymphocytic leukemia (right). The number of patients is shown for each figure. *p*-value represents the overall *p*-value of the graph, not comparisons between individual clusters.

**Table 1 ijms-24-05460-t001:** Summary overview of the RPPA arrays utilizing DNADR and DDR antibodies for acute and chronic leukemia samples.

Array Name	Disease	Age Category	# Samples	# Antibodies	# DNADR and DDR Antibody
AML A3	AML	Adult	810	412	21
TH2	AML	Pediatric	500	296	19
TH3	T-cell ALL	Adult	69	321	20
		Pediatric	292	321	20
CLL	CLL	Adult	795	384	24

**Table 2 ijms-24-05460-t002:** Patient and disease characteristics across the five protein clusters in acute myeloid leukemia.

AML		Total	C1	C2	C3	C4	C5	*p*
Number	Count	100%	9%	2%	38%	43%	8%	
Gender	Female	44%	50%	43%	46%	44%	36%	0.321
Age (years old)	Median	51.59	19.06	64.22	30.46	58.75	64.95	*p* < 0.001
	<2	5%	10%	0%	1%	7%	3%	*p* < 0.001
	2–10	13%	14%	11%	16%	11%	3%	0.005
	10–18	18%	25%	0%	25%	12%	10%	*p* < 0.001
	18–30	6%	8%	14%	8%	4%	0%	0.001
	30–60	18%	17%	14%	18%	18%	25%	0.404
	60+	41%	25%	61%	32%	48%	59%	*p* < 0.001
Ethnicity (*n* = 562)	Hispanic or Latino	13%	14%	7%	16%	11%	7%	0.859
Central nerve system involvement (*n* = 1269)	Positive	16%	24%	0%	19%	14%	4%	*p* < 0.001
Cytogenetics (*n* = 1202)	t(8;21)	8%	1%	4%	11%	7%	5%	0.002
	Inv16	7%	10%	0%	7%	7%	1%	0.043
	Diploid	30%	30%	18%	38%	25%	25%	*p* < 0.001
	MLL	9%	13%	7%	6%	11%	6%	0.022
	8, −5, −7	23%	18%	25%	16%	28%	30%	*p* < 0.001
	Other	16%	18%	21%	17%	14%	13%	0.340
Risk group (*n* = 1223)	Favorable	19%	18%	11%	26%	16%	7%	<0.001
	Intermediate	46%	50%	39%	45%	46%	44%	0.814
	Unfavorable	28%	24%	32%	25%	31%	33%	0.098
CEBPA mutation (*n* = 1068)	Mutated	8%	6%	4%	13%	5%	5%	0.001
NPM1 mutation (*n* = 1120)	Mutated	13%	9%	11%	20%	9%	4%	*p* < 0.001
FLT3-ITD mutation (*n* = 1133)	Mutated	16%	20%	7%	26%	9%	3%	*p* < 0.001
White blood cell count	>100,000	13%	31%	8%	12%	13%	1%	*p* < 0.001

**Table 3 ijms-24-05460-t003:** Patient and disease characteristics across the five protein clusters in T-cell acute lymphoblastic leukemia patients.

T-ALL		Total	C1	C2	C3	C4	C5	*p*
Number	Count	100%	48%	1%	25%	24%	2%	
Gender	Female	23%	23%	0%	17%	31%	14%	0.227
Age (years old at time of diagnosis)	Median		13	7.5	11	11	7	0.121
	<2	4%	4%	0%	2%	7%	0%	0.173
	2–10	37%	31%	100%	41%	41%	57%	
	10–18	33%	35%	0%	37%	29%	0%	
	18–30	15%	18%	0%	9%	15%	14%	
	30–60	9%	10%	0%	9%	8%	14%	
	60+	2%	2%	0%	2%	0%	14%	
Ethnicity (*n* = 338)	Hispanic or Latino	22%	26%	0%	20%	14%	43%	0.050
Race (*n* = 323)	Black	9%	8%	0%	13%	9%	0%	0.555
Central nerve system involvement (*n* = 349)	Positive	29%	34%	0%	28%	20%	29%	0.153
Early T-cell precursor (*n* = 342)	Yes	14%	13%	0%	20%	8%	14%	0.305
Risk group (*n* = 275)	Standard risk	24%	24%	0%	22%	28%	14%	0.689
	Intermediate risk	48%	45%	100%	55%	45%	43%	
	Very high risk	4%	2%	0%	7%	6%	0%	
T-cell receptor rearrangement (*n* = 312)	Yes	12%	10%	0%	10%	17%	14%	0.247
White blood cell count (*n* = 358)	>100,000	46%	51%	100%	51%	28%	43%	0.003

**Table 4 ijms-24-05460-t004:** Patient and disease characteristics across the five protein clusters in chronic lymphocytic leukemia patients.

CLL		Total	C2	Not C2	*p*
Number	Count	100%	96%	4%	
Age (years old at time of diagnosis)	Median	57	57	58	0.121
	30–60	57%	58%	55%	0.948
	60+	43%	42%	45%	
Gender	Female	39%	39%	38%	1.000
Race (*n* = 750)	Black	4%	4%	10%	0.234
	Not Black	90%	90%	86%	
	NA	6%	6%	3%	
Binet stage (*n* = 784)	A	62%	62%	52%	0.563
	B	9%	10%	7%	
	C	28%	27%	34%	
IGHV gene mutation status (*n* = 576)	Mutated	37%	38%	24%	0.269
Rai stage (*n* = 784)	0	34%	34%	28%	0.692
	I	29%	30%	21%	
	II	6%	6%	7%	
	III	17%	16%	24%	
	IV	13%	13%	14%	
Deletion 11q (*n* = 711)	Yes	13%	13%	7%	0.240
Deletion 13q (*n* = 711)	Yes	34%	35%	17%	0.027
Trisomy 12 (*n* = 711)	Yes	14%	14%	17%	0.413
Deletion 17p (*n* = 711)	Yes	9%	8%	24%	0.006
Chromosome 9 (*n* = 711)	Yes	2%	2%	10%	0.003

## Data Availability

Not applicable.

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
