# Peer review of "Reverse Phase Protein Array Profiling Identifies Recurrent Protein Expression Patterns of DNA Damage-Related Proteins across Acute and Chronic Leukemia: Samples from Adults and the Children’s Oncology Group"

_ijms, 2023, doi:10.3390/ijms24065460_

Round 1
Reviewer 1 Report
nice paper. it should be published
Author Response
Reviewer 1: nice paper. it should be published
Thank you for this generous feedback.
Reviewer 2 Report
The manuscript by Fieke W. Hoff et.al, demonstrated a new insight that recurrent protein expression patterns of DNA damage related proteins across acute and chronic Leukemia. They found that protein clusters showed similar implications for survival and remission duration in pediatric and adult T-ALL and AML populations, with C5 doing best in all. Collectively, DDR protein expression was abnormal in leukemia, formed recurrent clusters that were shared across the leukemias, with shared prognostic implications across diseases, and individual proteins showed age and diseases related differences. However, I have several issues to address before accepting publication.
Specific comment:
1. Line42-43: this part describe the multiple protein components affect the functionality of the various pathways controlling DDR. ‘Some of these genes are mutated in some forms of cancer, but within leukemia these genes are rarely mutated’ is unrelated with DDR.
2. Line52-56: the paper can talk about why CLL only detects adult population, but not pediatric population. ‘other associated small B cell leukemia/lysophomas’ should be deleted.
3. More descriptions of AML, A-TLL and CLL should be added to the introduction section.
4. Line64-65 it is better to delete the words ‘accompanying’.
5. Line68-70: the literature should be supplemented to support the computational methodology to recognize recurrent patterns of expression of all studied proteins researched by your group.
6. In figure 2, there is no explain to describe the C.
7. Line137-139: although there were no significantly differentially expressed proteins between patients < 60 years of ages compared to these older than 60 at time of diagnosis, it had better to provide supplementary figure to make the result more convincing.
8. To enhance the results, some functional verifications on these proteins connected to DDR could be added in this study.
9. To add depth to the debate, the rationale behind choosing these 16 proteins and their effects on AML, T-ALL, and CLL can be discussed.
Author Response
Reviewer 2:
Line 42-43: this part describes the multiple protein components affect the functionality of the various pathways controlling DDR. ‘Some of these genes are mutated in some forms of cancer, but within leukemia these genes are rarely mutated’ is unrelated with DDR.
We have added a statement that none of the genes are mutated in more than 1% of the leukemia cases.
Line 52-56: the paper can talk about why CLL only detects adult population, but not pediatric population. ‘other associated small B cell leukemia/ lymphomas’ should be deleted.
We have removed the statement about other associated small B cell leukemia/ lymphomas. We feel that we do not have to justify why we only included adult CLL patient, given that CLL is a disease of the older population.
More descriptions of AML, T-ALL and CLL should be added to the introduction section.
We have addressed this as suggested. We think that adding more information is beyond the scope of this article.
Line 64-65 it is better to delete the words ‘accompanying’.
We have addressed this accordingly.
Line 68-70: the literature should be supplemented to support the computational methodology to recognize recurrent patterns of expression of all studied proteins researched by your group.
We do not completely understand what kind of literature we should add to support the computational methodology. The used literature is cited. Please let us know how we can address this comment and we are happy to do so.
In figure 2, there is no explain to describe the C.
We have addressed this accordingly.
Line 137-139: although there were no significantly differentially expressed proteins between patients < 60 years of ages compared to these older than 60 at time of diagnosis, it had better to provide supplementary figure to make the result more convincing.
We have added a supplemental figure S1. RAD51 has a p-value of 0.044, which is not significant after correction for multiple comparisons.
To enhance the results, some functional verifications on these proteins connected to DDR could be added in this study.
While we agree that functional verifications would improve the strength of the paper, we are not able to perform wet lab experiments within the 5-day allotted time for the revisions.
To add depth to the debate, the rationale behind choosing these 16 proteins and their effects on AML, T-ALL, and CLL can be discussed.
We have elaborated on the rationale behind the selection of the antibodies as suggested.
Reviewer 3 Report
The authors have used a very reasonable cohort of acute myeloid (AML) (n=1310), T-cell acute lymphoblastic leukemia (T-ALL) (n=361), and chronic lymphocytic leukemia (CLL) (n=795) cases. Using clustering analysis study found 5 protein clusters. Three were found to be unique compared to normal CD34+ cells. abnormal expression of DDR protein in leukemia formed recurrent clusters that were shared across the leukemias, with shared prognostic implications across diseases, and individual proteins showed age and diseases related differences. This is an interesting study with clinical implications for the selection of therapy directed at DDR targets.
Minor comments:
The quality of Figures 2,3 and 4 needs to improve. It is difficult to read.
Author Response
Reviewer 3: The quality of Figures 2,3 and 4 needs to improve. It is difficult to read.
We have increased the font size of Figure 2, 3 and 4.